# Emerging Role of Circular RNAs in Hepatocellular Carcinoma Immunotherapy

**DOI:** 10.3390/ijms242216484

**Published:** 2023-11-18

**Authors:** Tasneem Abaza, Mostafa K. Abd El-Aziz, Kerolos Ashraf Daniel, Paraskevi Karousi, Maria Papatsirou, Sherif Ashraf Fahmy, Nadia M. Hamdy, Christos K. Kontos, Rana A. Youness

**Affiliations:** 1Biology and Biochemistry Department, Molecular Genetics Research Team (MGRT), Faculty of Biotechnology, German International University (GIU), Cairo 11835, Egypt; tasneem.abaza11@gmail.com (T.A.); mostafakhaled8899@gmail.com (M.K.A.E.-A.); kerolos.daniel@gaf.ac (K.A.D.); 2Biotechnology and Biomolecular Chemistry Program, Faculty of Science, Cairo University, Giza 12613, Egypt; 3Biochemistry Department, Faculty of Pharmacy, Al-Azhar University, Assiut Branch, Assiut 71631, Egypt; 4Biology and Biochemistry Department, Molecular Genetics Research Team (MGRT), School of Life and Medical Sciences, University of Hertfordshire Hosted by Global Academic Foundation, Cairo 11835, Egypt; 5Department of Biochemistry and Molecular Biology, Faculty of Biology, National and Kapodistrian University of Athens, 15701 Athens, Greece; pkarousi@biol.uoa.gr (P.K.); papatsir@biol.uoa.gr (M.P.); 6Department of Chemistry, School of Life and Medical Sciences, University of Hertfordshire Hosted by Global Academic Foundation, R5 New Garden City, New Capital, Cairo 11835, Egypt; sheriffahmy@aucegypt.edu; 7Biochemistry Department, Faculty of Pharmacy, Ain Shams University, Cairo 11566, Egypt; nadia_hamdy@pharma.asu.edu.eg

**Keywords:** circular RNAs (circRNAs), hepatocellular carcinoma (HCC), immunotherapy, cytotoxic T lymphocytes, natural killer cells, tumor microenvironment (TME)

## Abstract

Hepatocellular carcinoma (HCC) is a highly fatal malignancy with limited therapeutic options and high recurrence rates. Recently, immunotherapeutic agents such as immune checkpoint inhibitors (ICIs) have emerged as a new paradigm shift in oncology. ICIs, such as programmed cell death protein 1 (PD-1) inhibitors, have provided a new source of hope for patients with advanced HCC. Yet, the eligibility criteria of HCC patients for ICIs are still a missing piece in the puzzle. Circular RNAs (circRNAs) have recently emerged as a new class of non-coding RNAs that play a fundamental role in cancer pathogenesis. Structurally, circRNAs are resistant to exonucleolytic degradation and have a longer half-life than their linear counterparts. Functionally, circRNAs possess the capability to influence various facets of the tumor microenvironment, especially at the HCC tumor–immune synapse. Notably, circRNAs have been observed to control the expression of immune checkpoint molecules within tumor cells, potentially impeding the therapeutic effectiveness of ICIs. Therefore, this renders them potential cancer-immune biomarkers for diagnosis, prognosis, and therapeutic regimen determinants. In this review, the authors shed light on the structure and functional roles of circRNAs and, most importantly, highlight the promising roles of circRNAs in HCC immunomodulation and their potential as promising biomarkers and immunotherapeutic regimen determinants.

## 1. Introduction

Liver cancer ranks sixth in primary neoplasm prevalence. In comparison to other malignancies, liver cancer has an overwhelming impact marked by a high fatality rate. Global statistics expect liver cancer incidence to exceed one million cases by 2025 [1]. Hepatocellular carcinoma (HCC) predominates, making up 80–90% of primary liver malignancies [2], with a bleak five-year survival rate of 18% [3].

HCC management is intricate due to its diverse presentation in an inflamed liver, often compelling multidisciplinary teams to cooperate [4]. Surgical options offer a cure for early HCC; however, <20% of the patients receive a timely diagnosis [5]. Advanced-stage survival is recorded to be <10% in five years [6]. Unfortunately, more than 70% of HCC patients experience recurrence post-curative therapy [7,8]. Chemotherapy remained the primary form of HCC treatment despite all its complications, including chemoresistance and detrimental overall effects on the patients [9,10]. Yet, recent advances in understanding the molecular drivers of HCC have led to a new era of molecular targeting agents such as tyrosine kinase inhibitors (TKIs) and mammalian target of rapamycin (MTOR) inhibitors [11,12]. Nonetheless, TKIs have limited survival gains and associated intolerance, prompting innovative strategies.

In the field of oncology, immune checkpoint inhibitors (ICIs) are considered to be a novel and innovative strategy [13,14,15,16,17]. Their implementation has saved a lot of late-stage patients, especially patients with advanced unresectable HCC tumors [18,19]. ICIs brought a fundamental change, capitalizing on HCC’s immune pathogenicity [20]. Recent progress in understanding tumor–immune dynamics emphasizes ICIs targeting cytotoxic T-lymphocyte-associated protein 4 (CTLA-4) and programmed cell death protein 1 (PDCD1, also known as PD-1), which enhance therapeutic outcomes across several malignancies [16,21].

In 2017, nivolumab, a PD-1 inhibitor, was the first Food and Drug Administration (FDA)-approved monoclonal antibody (mAb) for second-line targeted therapy in the treatment of HCC. A year later, pembrolizumab, another PD-1 inhibitor, was added to the list. In 2020, the FDA approved two important combinational mAbs as immunotherapeutic regimens for HCC patients, which were nivolumab/ipilimumab and atezolizumab/bevacizumab [22,23]. In October 2022, tremelimumab/durvalumab became the most recently FDA-approved combinational mAb immunotherapeutic regimen for adult patients with advanced unresectable HCC [24]. It is worth noting that, nowadays, internationally endorsed guidelines adopt atezolizumab/bevacizumab as the first-line therapy for advanced, treatment-naïve HCC patients [24,25]. Recent studies (IMbrave150 trial) underscore atezolizumab/bevacizumab superiority over sorafenib in overall survival, progression-free survival, and patient-reported outcomes [26]. Although the impact of ICIs on survival is significant, they have also been linked to autoimmune-like side effects due to their ability to stimulate the immune system. These adverse effects are often expressed in the form of neurological toxicities, hepatotoxicity, and cardiotoxicity [27,28,29]. Therefore, we need a better understanding of the molecular mechanisms underlying therapeutic response.

The regulation of ICI pathways mediated by non-coding RNAs (ncRNAs) is a promising field of research regarding the probability of ICI toxicity. A substantial portion of the human genome undergoes transcription, yielding a diverse array of ncRNAs [30,31,32]. Within this spectrum, this discourse focuses on three key types—long non-coding RNAs (lncRNAs) [33,34], microRNAs (miRNAs) [35,36], and circular RNAs (circRNAs) [37]—that play fundamental roles in cancer pathogenesis [38,39,40].

In this review, the authors will focus on the most recently identified class of ncRNAs, which is circRNAs. circRNAs are closed-loop RNA structures, formed via the back splicing of precursor mRNA (pre-mRNA) molecules [41]. They are widely expressed in mammalian cells and known for stability, evolutionary conservation, and cell/tissue specificity. CircRNAs have diverse biological roles, including miRNA regulation, gene transcription modulation, RNA-binding protein (RBP) interaction, and protein/peptide encoding [42]. These functions primarily operate at epigenetic, transcriptional, and post-transcriptional levels [43,44]. CircRNAs regulate gene expression via an extended array of molecular mechanisms, influencing tumorigenesis and neoplastic progression [41]. Dysregulated circRNAs play pivotal roles in diseases, particularly in tumor development, influencing cell proliferation, apoptosis, and metastasis [45,46,47]. Most importantly, circRNAs have emerged as potent modulators of the tumor microenvironment (TME) and have a prospective role in tuning immunotherapeutic regimens’ efficiency and outcomes [16,39,48,49,50].

Coherently, the convergence of preclinical investigations consistently underscores the potential of manipulating ncRNAs to significantly potentiate the efficacy of immunotherapeutic interventions in the context of HCC. This scholarly review summarizes recent advancements in the landscape of circRNAs, followed by an in-depth exploration of the prospect of employing immunomodulatory circRNAs as plausible therapeutic targets/agents in HCC, accompanied by a comprehensive analysis of the intricate mechanistic frameworks that underlie these interactions.

## 2. Circular RNAs (CircRNAs)

### 2.1. What Are CircRNAs?

CircRNAs are a recently discovered class of ncRNA molecules. They are formed during the process of RNA transcript maturation. Structurally, circRNAs are covalently closed by a connection between a downstream donor and upstream acceptor RNA splice sites linked by a phosphodiester bond. CircRNAs were previously regarded as splicing junk but are now recognized as functional RNA molecules [31]. They have expression patterns that are particular to different tissues and cell types, and they are produced from a wide variety of genes [51]. It is noteworthy that circRNAs are implicated in biological processes that contribute to the development and spread of cancer [52,53].

Additionally, due to their circular shape and resistance to exoribonuclease activity, circRNAs have longer half-lives than their parental linear counterparts, making it possible to detect them even when produced at low levels [41,54,55]. For instance, exonic circRNAs are thought to be extremely stable in cells, with most circRNAs showing half-lives of over 48 h, as opposed to an average mRNA half-life of 10 h [54,56]. These characteristics imply that circRNAs could serve as useful biomarkers for the diagnosis and prognosis of cancer patients, as previously described in [41,50]. Yet, the current review focuses on the potential roles of circRNAs in modulating the HCC immunological profile and, thus, tuning the immune-suppressive TME in such chemo- and immuno-resistant tumors.

### 2.2. Biogenesis of CircRNAs

Recent research has shown that “back-splicing”, a type of pre-mRNA splicing, is responsible for the production of circRNAs [57]. CircRNAs have a distinctive closed-loop structure, created by linking a downstream 5′ splice donor site and an upstream 3′ splice acceptor site, in contrast to conventional pre-mRNAs with 5′ caps and 3′ polyadenylated tails [58,59].

CircRNAs are primarily categorized into four types [60,61] based on the origin of their genomic regions: exonic circRNAs (EcircRNAs), retained-intron or exonic-intronic circRNAs (EIcircRNAs), intronic circRNAs (ciRNAs), and tRNA intronic circRNAs (tricRNAs) [62]. Over 80% of the circRNAs that have been discovered are EcircRNAs, and these circRNAs are mostly found in the cytoplasm [63,64]. As EcircRNAs sponge miRNAs and/or interact with RBPs, several studies have shown that EcircRNAs play significant roles in modulating the genetic expression of several coding transcripts [65,66]. EIciRNAs and ciRNAs, which compose a minor portion of circRNAs compared to EcircRNAs, are mostly found in the nucleus and, thus, can control the expression of their parental mRNAs, as shown in Figure 1 [57]. The following section will cover the four associated biogenesis mechanisms of circRNAs.

#### 2.2.1. Intron Pairing-Driven Circularization

The most frequent circularization process of EcircRNA and EIciRNA involves “direct back-splicing”, also known as intron-pairing-driven circularization, in which a particular pre-mRNA with ALU repeats is sheared to generate an EcircRNA or an EIciRNA following reverse-base complementary pairing [56].

#### 2.2.2. RBP-Induced Circularization

RBPs, which are thought to be trans-acting factors and include Quaking (QKI), Muscleblind (MBL), and Fused-in Sarcoma (FUS), may promote circularization by bridging similar intronic sequences [67]. The 3′ and 5′ termini of circularized exons can be brought into closer proximity through the dimerization of RBPs. This dimerization process also facilitates splicing by engaging with the sequences both upstream and downstream of the circularized exons [68].

#### 2.2.3. Lariat-Induced Circularization Driven by Spliceosomes

Lariat-driven circularization, also known as the exon-skipping mechanism, occurs as pre-mRNA partially folds during transcription. This folding brings the 5′ splice site (donor site) of the upstream intron close to the 3′ splice site (receptor site) of the downstream intron, forming a circRNA through back-splicing within the folded region. The remaining exons then combine to create a linear mRNA [56]. Moreover, back splicing can occur post-transcriptionally or co-transcriptionally, involving either a single exon or multiple exons with intervening introns [69].

#### 2.2.4. Self-Circularization of Introns

Intron self-circularization occurs when a pre-RNA contains 7 nucleotides (nt) of guanine (G) and uracil (U)-rich sequence close to one exon and an 11 nt cytosine (C)-rich sequence close to another exon. This allows the introns to avoid branching and degradation during splicing, resulting in a stable intronic lariat structure [57].

### 2.3. Functional Roles of CircRNAs

CircRNAs typically function as regulatory ncRNA molecules, either directly by controlling gene transcription or indirectly by modifying other regulators, such as proteins and miRNAs. Further, the term “regulatory coding RNAs” refers to a subset of circRNAs that encode short functional peptides, as shown in Figure 2 and described below [53].

#### 2.3.1. miRNA Sponge

Some circRNAs may behave as miRNA sponges or sequesters because they include well-conserved canonical miRNA response elements (MREs) [70,71,72]. Some circRNAs that act as miRNA sponges can positively or adversely affect the expression of the corresponding targeted genes. Cerebellar degeneration-related protein 1 antisense (CDR1-AS or ciRS-7), a well-studied circRNA, has been linked to a variety of malignancies, including HCC and gastric cancer, as well as sponges miR-7 in embryonic zebrafish [73,74,75]. Indeed, a growing body of research has shown that the circRNA-miRNA-mRNA regulatory network may have significant effects on several diseases, including HCC [76,77].

For instance, circ-ZNF609 increases the expression of the myocyte-specific enhancer factor 2A (MEF2A), which improves vascular endothelial dysfunction by acting as an endogenous miR-615-5p sponge to decrease miR-615-5p activity [78]. Our previous work has also validated the tumor suppressor and immunomodulatory effects of miR-615-5p in HCC cell lines and primary natural killer (NK) cells isolated from HCC patients [79]. Thus, we highlighted that the potential activity of circ-ZNF609 in HCC patients deserves further investigation.

According to Zhong et al., circ-MYLK can ease the inhibition of its target vascular endothelial growth factor A (VEGFA), a crucial component of the VEGFA/VEGFR2/RAS/MAPK1 signaling pathway, in addition to being associated with the stage and grade of bladder carcinoma [80]. By sequestering miR-143 and increasing the production of its target *BCL2*, increased levels of circ-UBAP2 stimulate the proliferation of osteosarcoma cells while preventing apoptosis both in vitro and in vivo [81]. Similarly, circ-ABCB10 has been shown by Liang et al. to sponge miR-1271, promoting proliferation and inhibiting the apoptosis of breast cancer cells [82].

#### 2.3.2. Protein Sponge or Decoy

CircRNAs can also bind and sequester proteins using their protein-binding sites, functioning as an antagonist to impede their physiological function [83]. RBPs are one of the most common protein classes that can bind to circRNAs. For instance, circ-TNPO3 functions as a protein decoy for the insulin-like growth factor 2 mRNA binding protein 3 (IGF2BP3) to inhibit the capacity of gastric cancer cells to proliferate [84]. It is also worth noting that our previous work has highlighted the potent role of IGF2BPs in regulating HCC tumor activity and their potential regulation with miRNAs, including miR-1275 [85]. The expression of MYC proto-oncogene, as well as bHLH transcription factor (MYC) and its target, snail family transcriptional repressor 1 (SNAI1), is inhibited when circ-TNPO3 sequesters IGF2BP3, which reduces the ability of gastric cancer cells to proliferate and metastasize [84]. It was also reported that circ-SIRT1 binds to the eukaryotic translation initiation factor 4A3 (EIF4A3) in colorectal cancer cell lines, preventing its inhibitory impact on epithelial–mesenchymal transition and encouraging the proliferation and invasion of colorectal cancer cell lines [86].

CircRNAs can also decoy proteins by attaching themselves to cellular proteins and changing how they normally carry out their physiological functions [44,87]. Circ0000079 (ciR79) inhibits the induction of fragile X-related 1 (FXR1) protein and prevents its complexation with protein kinase C iota (PRKCI), thus preventing the FXR1/PRKCI-mediated phosphorylation of glycogen synthesis kinase 3β (GSK3B) and activator protein 1 (AP-1), suppressing SNAI1 protein levels and hindering non-small cell lung cancer growth [88].

#### 2.3.3. Protein Scaffolding

CircRNAs with enzyme and substrate binding sites are believed to serve as scaffolds that help two or more proteins to come into proximity and interact. CircFoxo3, which includes binding sites for MDM2 and p53, serves as an indicative case of this observation. In order to support the idea that circFoxo3 can serve as a protein scaffold, the mutation of these binding sites or circRNA silencing reduced the amount of p53 that an MDM2 antibody could pull down. Further research revealed that circFoxo3 promoted the ubiquitination of p53 by MDM2, which is then destroyed by the proteasome. Additionally, circACC1 forms a ternary complex with the regulatory β and γ subunits of AMP-activated protein kinase (AMPK), stabilizing and enhancing the enzymatic activity of the AMPK holoenzyme [89]. More circRNAs acting as scaffolds are expected to be identified in the future because of the longer half-lives of circRNAs [90].

#### 2.3.4. Transcriptional Regulation

CircSEP3 derived from *SEP3* exon 6 enhances the abundance of homologous exon 6-skipped variant by attaching to the host DNA locus and creating an RNA-DNA hybrid or R-loop, which causes transcription to pause and splicing factor recruitment [91]. Similarly, circSMARCA5 induces the expression of the shortened non-functional isoform by causing transcriptional termination of the SWI/SNF-related, matrix-associated, and actin-dependent regulator of chromatin, subfamily a, member 5 (*SMARCA5)* at exon 15 through R-loop formation [92]. EIciRNAs can interact with the U1 small nuclear ribonucleoprotein to increase the expression of parental genes through RNA-RNA interactions with snRNA molecules [93]. As lariats evade debranching, circRNAs can amass at their formation sites and enhance the activity of RNA polymerase II, resulting in the increased expression of the respective genes [57].

#### 2.3.5. Translation to Proteins and Peptides

The ability of circRNAs to undergo translation was originally discovered by Pamudurti et al. in 2017 [94]. According to bioinformatics studies, some circRNAs contain an open reading frame (ORF), which indicates that they can be translated. Ribosome profiling, which can sequence ribosome-covered RNAs to track translation in vivo, has shown convincing evidence that some circRNAs comprising internal ribosome entry sites (IRES) are translated based on an IRES-dependent mechanism [94], whereas other circRNAs are translated independently of IRES elements. The translation of circSHPRH into the SNF2 histone linker PHD RING helicase (SHPRH)-146aa protein was demonstrated to be IRES-dependent. It was discovered that SHPRH-146aa is a tumor suppressor protein that guards against the degradation of the SHPRH full-length protein. Therefore, incorrect circSHPRH translation affects tumor malignancy [95].

Other circRNAs have also been discovered to encode functional peptides and proteins that have tumor-promoting or -suppressing properties [95,96,97]. Finally, certain circRNAs can encode peptides without the need for IRES. Protein translation is made easier, for instance, by the m^6^A modification. The m^6^A reader protein YTH N6-methyladenosine RNA binding protein F3 (YTHDF3) interacts with translation initiation factors to start protein synthesis by binding to circRNAs that include m^6^A modification sites [98,99,100].

#### 2.3.6. Regulation of Epigenetic Alterations

Cancer commonly exhibits abnormal DNA methylation and histone alterations that are linked to the epigenetic regulation of gene expression [101,102]. It has been discovered that certain circRNAs control these epigenetic changes. According to Chen et al. [103], circFECR1 significantly reduced the amount of CpG DNA methylation in the promoter of Fli-1 proto-oncogene, ETS transcription factor (FLI1), which epigenetically activated FLI1. Through binding to the DNA methyltransferase 1 (DNMT1) promoter, circFECR1 has been shown to suppress the transcription of DNMT1, a crucial methyltransferase enzyme necessary for the upkeep of DNA methylation. Additionally, tet methylcytosine dioxygenase 1 (TET1) DNA demethylase might be attracted by circFECR1 to the FLI1 promoter and cause DNA demethylation. A component of polycomb-repressive complex 2 (PRC2), as an enhancer of zeste homolog 2 (EZH2), serves as an H3K27 methyltransferase and controls histone methylation [104,105]. Moreover, hsa-circ0020123 can upregulate EZH2 and zinc finger E-box binding homeobox 1 (ZEB1) using sponging miR-144, while circBCRC4 can enhance the expression of EZH2 by interacting with miR-101 [106,107].

### 2.4. Involvement of circRNAs in HCC Tumor Development and Progression

It has been reported that circRNAs have a fundamental role in the etiology of several human diseases, including several oncological conditions [108]. According to earlier investigations, circRNAs are thought to be important to the onset, development, and growth of HCC. For instance, circ0008450 induces HCC cellular proliferation, invasion, and migration and reduces apoptosis caused by sponging miR-548 [109]. Additionally, circRNA-104718 can similarly enhance HCC cellular proliferation, invasion, and migration and inhibit apoptosis by regulating the miRNA-218-5p/TXNDC5 axis [110]. The circular RNA hsa_circ_0078710 enhances cell proliferation by sequestering miR-31, resulting in the upregulation of histone deacetylase 2 (HDAC2) and cyclin-dependent kinase 2 (CDK2) expression [111]. Circ-ZEB1.33 facilitates the proliferation of HCC cells by modulating the miR-200a-3p/CDK6axis [112]. Hsa_circ_0016788 expedites HCC growth through the regulation of miR-481 and its downstream target cyclin-dependent kinase 4 (CDK4) [113]. Furthermore, hsa_circ_0091581 promotes the proliferation of HCC cells by elevating MYC levels, acting as a sponge for miR-526b [114]. Additionally, circBACH1 directly interacts with the RNA binding protein HuR, promoting the cytoplasmic accumulation of HuR, thus leading to decreased cyclin-dependent kinase inhibitor 1B (CDKN1B) expression [115], which influences cell cycle progression.

Pu et al. observed a significant increase in hsa_circ_0000092 expression in HCC tissues and cell lines. Depleting hsa_circ_0000092 inhibited HCC cell proliferation, migration, invasion, and angiogenesis in vitro and in vivo. This circRNA promotes HCC angiogenesis by acting as a miR-338-3p sponge, leading to increased expression of Jupiter microtubule-associated homolog 1 (JPT1), matrix metallopeptidase 9 (MMP9), and VEGFA [116]

Recent research highlights the pivotal roles of circRNAs in the regulation of apoptotic mechanisms within HCC. Specifically, these circRNAs target key components involved in both anti-apoptotic and pro-apoptotic signaling pathways. Notably, circ-BIRC6 exhibits significant overexpression in HCC tissue samples and correlates with the overall survival of HCC patients. Silencing circ-BIRC6 expression effectively enhances apoptosis in HCC cells by modulating BCL2 apoptosis regulator (BCL2) levels through the sequestration of miR-3918 [117]. Moreover, circ-0051443 displays reduced expression in HCC tissues and plasma. Exosomal circ-0051443 exerts a suppressive influence on the biological behaviors of HCC cells, primarily by promoting apoptosis through the interaction with miR-331-3p and the regulation of BCL2 antagonist/killer 1 (BAK1) [118]. On the other hand, certain circRNAs have inhibitory influences on the development of HCC. For instance, circADAMTS14 regulates miR-572/RCAN1, leading to the abrogation of HCC cellular hallmarks and inducing HCC cellular apoptosis machinery [119]; circRNA-5692 has a similar inhibitory impact on HCC progression by controlling the miR-328-5p/DAB2IP axis [120]. Table 1 represents a comprehensive list of all characterized oncogenic and tumor suppressor circRNAs in HCC.

## 3. Immunotherapy

Cancer immunotherapeutic modalities include several strategies, such as chimeric antigen receptor (CAR) T-cells, tumor vaccines, oncolytic viruses, and ICIs [174]. CAR T-cell therapy is a type of treatment through which a patient’s T cells are modified in a laboratory to attack cancer cells. This is performed by adding a gene for a particular receptor called a CAR to the T cells [49]. The modified CAR T cells are then grown in large numbers and infused back into the patient to kill tumor cells [175].

Cancer vaccine activates the body’s anti-tumor defenses by introducing tumor antigens. These antigens can be administered in various forms, such as whole cells, peptides, or nucleic acids. An ideal cancer vaccine aims to counteract the immune suppression present in tumors and stimulate both humoral and cellular immunity [176].

Oncolytic viruses are used as therapeutic agents to stimulate the selective destruction of tumor cells, allowing the targeted eradication of tumor cells while leaving normal tissues unaffected, and triggering anti-tumor immunity [177].

ICIs such as anti-CTLA4 and anti-PD-1 antibodies target immune checkpoint molecules present on immune cells to inhibit their activities, as previously described, thereby alleviating immunosuppression and prompting CD8+ T cells to eliminate cancerous cells in the body [178].

While the clinical outcomes of ICI appear to be promising, the overall level of response remains inadequate, as only 20% of individuals with solid tumors experience complete remission (CR) following treatment [179]. Such recent advancements in cancer immunotherapy and its combination regimens have greatly affected the HCC treatment outcomes, and clinical studies are continuing to pave the way for leveraging the additional benefits for HCC patients.

### 3.1. HCC Immunotherapy

HCC patients who solely depend on surgery, chemotherapy, or radiotherapy not only have low chances of survival but also do not experience significant improvements in their quality of life [12,18,85,180]. However, the introduction and rapid advancements in cancer immunotherapy, as described earlier, have provided increasingly promising results [79,181]. The presence of immune cells within the TME is crucial to fighting against tumors. However, cancerous cells can avoid the immune system and establish a complex equilibrium wherein diverse types of immune cells may contribute to the advancement of the tumor, spread to other parts of the body, and show resistance to treatment. Novel immunotherapy strategies are focused on reinstating the original equilibrium and enhancing the immune response against cancer through various means [182].

Immunotherapeutic regimens have successfully increased the survival rates, minimizing the side effects and providing long-term cancer control in advanced unresectable HCC patients [183]. Yet, a reliable marker for selecting the patients who would benefit the most from HCC immunotherapeutic regimens and those who would exhibit severe side effects is still missing [31,47].

Cell-free or circulating nucleic acids (CNAs) such as circRNAs in the blood have recently been identified as a new class of promising cancer–immune diagnostic/prognostic biomarkers to achieve the best outcome in HCC patients [31,184]. Prostate cancer-associated 3 (PCA3) has, in particular, been approved by the FDA and is currently being sold as Progensa by Hologic Gen-Probe (Marlborough, MA, USA) for prostate cancer diagnosis [185]. Circulating ncRNAs such as PCA3 are more reliable than other CNAs due to their high stability in the bloodstream and resistance to nuclease-mediated fragmentation, as extensively studied and reviewed by our research group in [186,187,188,189,190,191]. Plasma lncRNAs, in particular, were reported to be less sensitive to degradation induced via repetitive freeze–thaw cycles, as well as prolonged exposure to 45 °C and room temperature [192]. In this section of the review, the authors will shed light on the roles of circRNAs in HCC immunotherapy.

### 3.2. CircRNAs in HCC Immunotherapy

Since circRNAs have the potential to regulate many aspects of tumor immunity, they play a significant role in tumor immunotherapy; circRNAs have been observed to regulate the expression of immune checkpoint molecules in tumor cells, thereby allowing circRNAs to potentially hamper the therapeutic efficacy of ICIs [193]. Blocking the PD-1/PD-L1 checkpoint is one of the immunotherapeutic tactics widely used for treating various tumor types, including HCC [194].

CircRNAs in HCC can trigger immune system suppression and result in resistance to anti-PD-1 therapy. Evidence suggests that certain circRNAs can induce immune suppression and resistance against anti-PD-1 therapies in HCC. An illustrative case is circMET, an oncogenic immunosuppressor circRNA that triggers immune suppression through the SNAI1/DPP4/CXCL10 axis. Notably, sitagliptin, a dipeptidyl peptidase 4 (DPP4) inhibitor, augments CD8+ T cell infiltration in HCC tissues in diabetic individuals, potentially enhancing PD-1 blockade-based immunotherapy in selected HCC patients [195]. Therefore, the use of a DPP4 inhibitor or circMET siRNAs may significantly improve the effectiveness of immunotherapy with PD-1 blockade for the studied group of HCC patients.

CircPRDM4 is another immune-suppressor circRNA that works via a direct modulatory effect on the immune checkpoint, namely PD-L1 expression on HCC cells. CircPRDM4 induces the elevation of PD-L1 expression and facilitates the recruitment of HIF-1α onto the CD274 promoter under hypoxic conditions, resulting in CD8+ T cell-mediated immune evasion, as shown in Figure 3 [196]. This suggests that circPRDM4 is a promising onco-immune target in HCC. The presence of circRHBDD1 hinders the effectiveness of anti-PD-1 therapy in individuals with HCC. In HCC patients who respond to anti-PD-1 treatment, circRHBDD1 is found to be significantly elevated. However, when circRHBDD1 is targeted, the efficiency of anti-PD-1 therapy is enhanced in an immune-competent mouse model [197].

Additionally, it was reported that the overexpression of exosomal circTMEM181 secreted by tumor cells could impede the effectiveness of anti-PD-1 therapy in HCC and promote immunosuppression by upregulating the expression of CD39. Additionally, it hinders the ATP-adenosine pathway by targeting CD39 on macrophages. Conversely, rescuing the resistance to anti-PD-1 therapy in HCC can be achieved by targeting CD39 on macrophages [198]. Lastly, HCC-derived circCCAR1 was also reported to have an unfavorable effect on TME by inducing the permeant cellular dysfunction of CD8+ T-cells, and it is, thus, considered one of the most deleterious immune-evasion tactics orchestrated by HCC tumors (Figure 3). CircCCAR1 also plays a role in causing resistance to anti-PD-1 immunotherapy, suggesting a potential onco-immune target for HCC patients [199].

Regulatory T cells have the potential to disrupt the immune microenvironment of tumors and encourage immune evasion by suppressing the activation of effector T cells, such as CD4+ and CD8+ T cells [200,201]. Huang et al. conducted a study revealing that T cells can take up exosomes containing circGSE1. This uptake process plays a crucial role in enhancing the differentiation of CD4+ T cells into regulatory T cells by activating the miR-324-5p/TGFB1/SMAD3 pathway. The expansion of regulatory T cells, in turn, contributes to the increased proliferation, migration, and invasion of HCC. Consequently, exosomal circGSE1 holds promising potential as a target for immunotherapy [202].

NK cells play a crucial role in HCC TME, from the adaptive immune to the innate immune arm. Thus, the augmentation of NK cell infiltration and functionality increases patient survival across the spectrum of HCC patients [18]. This finding provides a novel vantage point for manipulating NK cell activity to augment the responsiveness of immunotherapeutic regimens among HCC patients [79,203]. Noteworthy investigations have unveiled instances of NK cell dysfunction within the HCC context, although the precise underpinnings of this phenomenon remain ambiguous. Zhang and collaborators have proffered insights into the mechanistic terrain, delineating how circUHRF1 orchestrates HCC progression and immune repression via an exosome-mediated and NK cell-dependent modality [127]. Mechanistically, circUHRF1 engenders NK cell dysfunction by sequestering miR-449c-5p, thus fostering hepatitis A virus cellular receptor 2 (HAVCR2, also known as TIM-3) expression. Significantly, circUHRF1 potentiates resistance to anti-PD-1 immunotherapy. Therefore, the strategic targeting of circUHRF1 emerges as a promising path for enhancing the therapeutic potency of anti-PD-1 immunomodulation in the HCC domain [127]. Moreover, Ma et al. utilized a particular plasmid to induce the overexpression of circARSP91 and then investigated how HCC cells would respond to NK cell cytotoxicity. They discovered that the UL16 binding protein 1 (ULBP1) showed a increased expression level, suggesting its potential influence on activating NK cells. Ultimately, their findings led them to conclude that circARSP91 can enhance the cytotoxicity of NK cells acting against tumors by upregulating ULBP1 (Figure 3) [204].

## 4. Conclusions and Future Perspectives

In conclusion, this review focuses on circRNAs as a novel investigational course of treatment in the field of tumor immunotherapy, highlighting the promising roles of circRNAs in the context of immunomodulation, which holds encouraging potential for improving the survival rates and outcomes of HCC patients. The authors shed light on the biogenesis of circRNAs, their functional modes, and their roles in HCC development and progression. However, special focus was given to summarizing the current literature discussing the roles of circRNAs as potential regulators of the immunogenic profile of HCC cells, cytotoxic T cells, and NK cells at the TME. Also, the review shed light on the promising roles of circRNAs as messengers between cancer and immune cells at the cancer–immune synapse at the TME. Yet, the authors also highlighted the gap in the literature concerning the mechanistic roles of circRNAs as possible modulators of other immune cells present at the TME, such as tumor-associated macrophages, tumor-associated fibroblasts, dendritic cells, and T regulatory cells. Nonetheless, the roles of circRNAs in regulating the immune-suppressive cytokine storm surrounding the tumor at the TME are under-investigated in HCC. This review paves the way for future possible usage of circRNAs to potentially offer a novel approach to enhance anti-tumor immune responses and overcome the challenges associated with current immunotherapeutic approaches available for HCC patients.

## Figures and Tables

**Figure 1 ijms-24-16484-f001:**
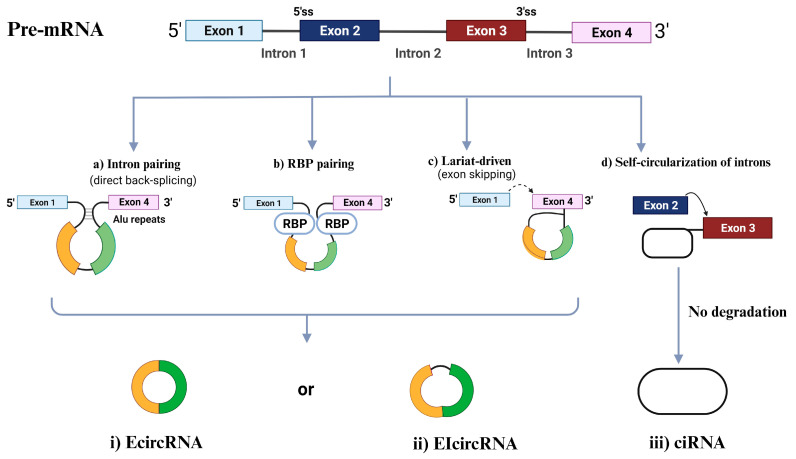
Circular RNA (circRNA) biogenesis. Schematic presentation of the different mechanisms of circRNAs biogenesis: intron painting (**a**), RBP pairing (**b**), Lariat driven (**c**), and the self-circularization of introns (**d**). ciRNA, intronic circRNA; EcircRNA, exonic circRNA; EIcircRNAs, exon-intron circRNA; RBP, RNA-binding protein; ss, splice site.

**Figure 2 ijms-24-16484-f002:**
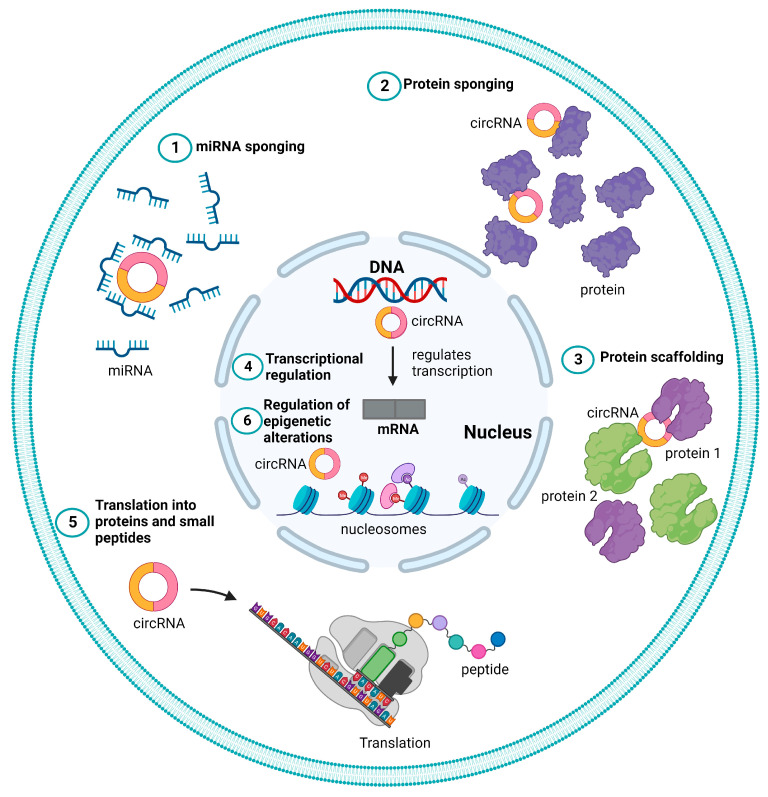
Different functional roles of circular RNAs (circRNAs). Schematic representation of different mechanisms of action of circRNAs represented as (1) microRNA (miRNA) sponge, (2) protein sponge or decoy (3) protein scaffolding, (4) transcriptional regulation, (5) translation to proteins, and peptide (6) regulation of epigenetic alterations.

**Figure 3 ijms-24-16484-f003:**
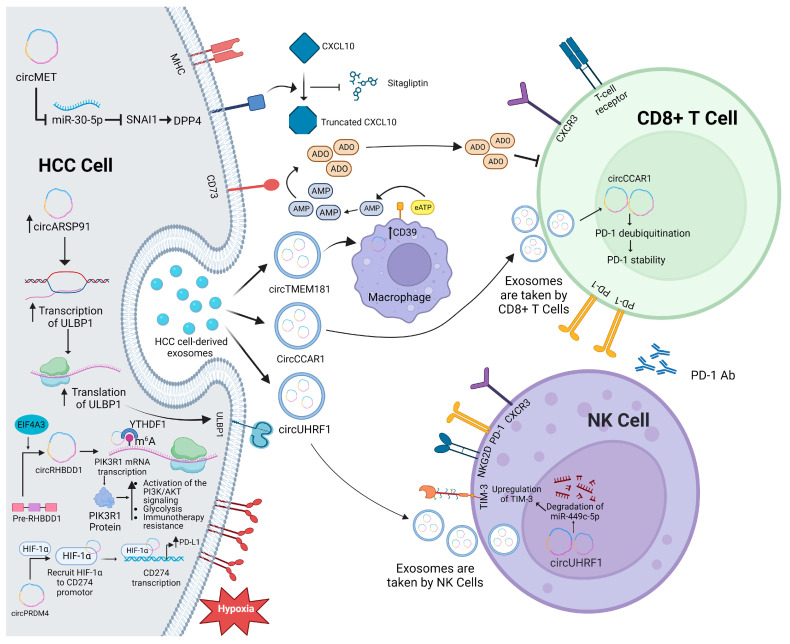
A snapshot of potential circRNAs as promising modulators of tumor microenvironment in HCC. Ab, antibody; ADO, adenosine; AKT, AKT serine/threonine kinase 1; AMP, adenosine monophosphate; CD39, ectonucleoside triphosphate diphosphohydrolase 1; CD73, 5′-nucleotidase ecto; CD8, cluster of differentiation 8; CXCL10, C-X-C motif chemokine ligand 10; CXCR3, C-X-C motif chemokine receptor 3; DPP4, dipeptidyl peptidase 4; eATP, extracellular adenosine triphosphate; EIF4A3, eukaryotic translation initiation factor 4A3; HCC, hepatocellular cancer; HIF-1a, hypoxia inducible factor 1 subunit alpha; M^6^A, N6-methyladenosine; MHC, major histocompatibility complex; NK, natural killer; NKG2D, killer cell lectin-like receptor K1; PD-1, programmed cell death protein 1; PD-L1, CD27/programmed cell death protein 1 ligand; PI3K, phosphoinositide 3-kinase; PIK3R1, phosphoinositide-3-kinase regulatory subunit 1; RHBDD1, rhomboid domain containing 1; Snail, snail family transcriptional repressor 1; TIM-3, hepatitis A virus cellular receptor 2; ULBP1, UL16 binding protein 1; YTHDF1, YTH N6-methyladenosine RNA binding protein F1.

**Table 1 ijms-24-16484-t001:** Oncogenic and tumor suppressor circular RNAs in hepatocellular carcinoma (HCC).

Circular RNA	Class	Molecular Targets	In Vitro/In Vivo/Ex Vivo Model	References
SCD-circRNA2	Oncogenic	MAPK1, RBM3	Huh7 HepG2 HCT-15 NCI-N87	[121]
circRHOT1	Oncogenic	NR2F6	HCC Tissues	[122]
circ-100338	Oncogenic	MMP2, MMP9	Hep3B HLE Huh7 BEL7402 SMCC7721 MHCC97L MHCC97H HCCLM3 HCCLM6	[123]
circ-0000092	Oncogenic	miR-338-3p	Hep3B LM3 MHCC97L SK-hep1 HepG2	[116]
circPRMT5	Oncogenic	miR-188-5p/HK2 axis	HCC tissues HCCLM3 SNU-387	[124]
circMAT2B	Oncogenic	PKM2	HepG2 Huh7 SMMC-772 MHCC-97L MHCC-97H	[125]
circASAP1	Oncogenic	MAPK1	MHCC97L MHCC97H HCCLM3	[126]
circβ-catenin	Oncogenic	β-catenin	Huh7	[97]
circUHRF1	Oncogenic	UHRF1	HepG2 HCCLM3 SMMC-7721 Huh 7 PLC/PRF/5 Hep3B	[127]
circ-CDYL	Oncogenic	PI3K-AKT-MTORC1/β-catenin and NOTCH2	HCCLM SMMC7721	[128,129]
circ-0046600	Oncogenic	HIF-1α	HepG2 SK-HEP-1	[130]
hsa_circ_0101432	Oncogenic	MAPK1	Huh-7 SK-HEP-1 HepG2 HLE	[131]
circMAN2B2	Oncogenic	MAPK1	HL-7702	[132]
circPTGR1	Oncogenic	MET	HepG2 97L LM3	[133]
circ-DB	Oncogenic	miR-34a, and USP7	HepG2 Hepa 1-6 3T3L1	[134]
circRNA Cdr1as	Oncogenic	AFP	SMMC-7721 Bel-7402 HepG2 Hep3B Huh-7 HB611	[135]
circRNA PVT1	Oncogenic	miR-203/HOXD3 pathway	SMMC-7721 Huh-7	[136]
circPVT1	Oncogenic	*TRIM23*/miR-377 axis	SNU-387 Huh-7	[137]
hsa_circ_0008450	Oncogenic	EZH2	SMMC7721 Sk-Hep-1 HepG2 Huh-7 HCCLM3	[138]
circ_0008450	Oncogenic	miR-548	HepG2 Huh-7, SMMC7721 Sk-Hep-1 HCCLM3	[109]
hsa_circRNA_103809	Oncogenic	miR-377-3p/FGFR1/MAPK1 axis	MHCC97L Huh7 SK-HEP-1 Hep3B HCCLM3	[139]
circRNA-104718	Oncogenic	miR-218-5p/TXNDC5	HCC nude mice model	[110]
circMYLK	Oncogenic	miR-362-3p/Rab23	Huh7 Hep3B	[140]
circ-ZNF652	Oncogenic	miR-29a-3p/GUCD1 Axis	SNU-387 Huh-7	[141]
circ_0000267	Oncogenic	miR-646	HepG2 Huh-7 SMMC7721 Sk-Hep-1 HCCLM3	[142]
circ-FOXP1	Oncogenic	miR-875-3p, miR-421, SOX9 factor	SNU-387 HepG2 Hep3B Huh7 SMMC-7721 HCCLM3	[143]
circRNA_104075	Oncogenic	YAP-dependent tumorigenesis through regulating HNF4a	Bel-7402 SMMC-7721 Huh7 HepG2 Hep1 Bel-7404 THLE-3 HL-7702	[144]
hsa_circ_101280	Oncogenic	miR-375/JAK2	HepG2 SNU-398	[145]
circRNA-101368	Oncogenic	HMGB1/RAGE	HCCLM3 HepG2	[146]
circ-ZEB1.33	Oncogenic	miR-200a-3p-CDK6	97H Huh7 HepG2 SNU423 SNU475 L02	[112]
circFBLIM1	Oncogenic	miR-346	HCC tissues HCC mouse model	[147]
hsa_circ_0103809	Oncogenic	miR-490-5p/SOX2 signaling pathway	MHCC97H HepG2 Huh7 SMMC7721 SK-Hep1	[148]
hsa_circ_0016788	Oncogenic	miR-486/CDK4	HepG2 Hep3B Huh7 HCCLM3 MHCC97L	[113]
hsa_circRBM23	Oncogenic	miR-138	HCC tissues HepG2 Huh7 Bel-7402	[149]
hsa_circ_0005075	Oncogenic	miR-431	SMMC-7721	[150]
circABCC2	Oncogenic	miR-665	HepG2 Bel-7402 MHCC97H	[151]
hsa_circ_100338	Oncogenic	MTOR signaling pathway	SMMC7721 Bel-7402 Hep3B	[152]
circ_0091581	Oncogenic	miR-591/FOSL2 axis	THLE-2	[153]
circPCNX	Oncogenic	miR-506	HL-7702 SMMC-7721 HuH-7 Hep3B HepG2	[154]
hsa_circ_0056836	Oncogenic	miR-766-3p/FOSL2 axis	Huh7 HepG2 SNU449 SK-HEP-1	[155]
circ- HOMER1	Oncogenic	miR-1322 on CXCL6	Sk-Hep-1 SMMC7721 HCCLM3 Huh-7 HepG2	[156]
circ_0091579	Oncogenic	miR-136-5p/TRIM27 miR-1270/YAP1 miR-1225/PLCB1	HCCLM3 MHCC97H Huh-7	[157,158,159]
circ_0001955	Oncogenic	miR-516a-5p miR-646 miR-145-5p/NRAS	Huh-7 HepG2 SMMC-7721 Bel-7402 Hep-3B	[160,161,162]
circTRIM33-12	Tumor suppressor	miR-191	HCC tissues MHCC97-L MHCC97-H LM3	[163]
circHIAT1	Tumor suppressor	PTEN	Hep3B SMMC-7721 HepG2 LM3	[164]
circLARP4	Tumor suppressor	miR-761/RUNX3/p53/CDKN1A pathway	Huh7 Hep3B SMMC7721 HepG2	[165]
circMTO1	Tumor suppressor	miR-9-5p/NOX4 axis	HepG2 Hep3B	[166]
circITCH	Tumor suppressor	miR-184	Huh7 HCCLM3 SMMC-7721 MHCC97H HepG2	[167]
circFBXW4	Tumor suppressor	miR-18b-3p/FBXW7 axis	LX-2	[168]
mmu_circ_34116	Tumor suppressor	miR-661/PTPN11	HepG2, SNU449	[169]
hsa_circ_0007874/cMTO1	Tumor suppressor	miR-338-5p	HCCLM3 MHCC97-L Hep3B SMMC-7721 Huh7 Bel-7402 MHCC97-H	[170]
circ608	Tumor suppressor	miR-222/PINK1	Primary hepatic stellate cells (PHSCs) from C57BL/6 mice	[171]
hsa_circ_0070963	Tumor suppressor	miR-223-3p LEMD3	LX2	[172]
hsa_circ_0004018	Tumor suppressor	miR-626/DKK3	Huh7 Bel7402 SNU182 Hep3B SNU449	[173]

## Data Availability

Not applicable.

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
