# Peer review of "Emerging Role of Circular RNAs in Hepatocellular Carcinoma Immunotherapy"

_ijms, 2023, doi:10.3390/ijms242216484_

Round 1

Reviewer 1 Report

Comments and Suggestions for Authors

The review from Mostafa K. Abd El-Aziz et al. is very well written and present a collection of recent advancements in the field of circRNAs, also highlighting any points that have remained unexplored and suggesting new research directions.

It would be interesting for the review to describe any differences in expression of the different circRNAs found in HCCs with different etiologies, but this is beyond the scope of the review

Author Response

ReVIEWER’S Comments and Corresponding Responses

Reviewer #1 (Comments to the Author):

  1. The review from Mostafa K. Abd El-Aziz et al. is very well written and present a collection of recent advancements in the field of circRNAs, also highlighting any points that have remained unexplored and suggesting new research directions. It would be interesting for the review to describe any differences in expression of the different circRNAs found in HCCs with different etiologies, but this is beyond the scope of the review.

We appreciate the Reviewer’s positive remarks on the manuscript. While we acknowledge the importance of investigating differences in circRNA expression related to HCC etiologies, the primary aim of this review is to provide a comprehensive overview of the current state of knowledge regarding the role of circRNAs in HCC immunotherapy, their functional roles, and their potential as biomarkers and immunotherapeutic regimen determinants. Therefore, we decided not to include this aspect in order to maintain the focus and clarity of our review.

The authors wish to thank the Reviewers for their constructive comments that led to the improvement of the current manuscript.

Reviewer 2 Report

Comments and Suggestions for Authors

Mostafa K. Abd El-Aziz et al have written a nice review presenting circRNAs as novel agents in the area of tumor immunotherapy and highlighting their promising role not only as immunomodulators but also as biomarkers and therapeutic alternatives for HCC patients. 

I would suggest to include a short sentence in the introduction, following line 79 to briefly describe the current problem with ICIs and also a linker to the non-coding RNAs paragraph, since the change from current FDA approved ICIs to the non-coding RNAs seems a bit abrupt. 

I would also suggest to choose either TIME or TME. 

  Comments on the Quality of English Language

Regarding the quality of English Language, I would suggest to tone down a little bit expressions such as "realm" "maestro" "sanguine"

Author Response

ReVIEWER’S Comments and Corresponding Responses

Reviewer #2 (Comments to the Author):

  1. I would suggest to include a short sentence in the introduction, following line 79 to briefly describe the current problem with ICIs and also a linker to the non-coding RNAs paragraph, since the change from current FDA approved ICIs to the non-coding RNAs seems a bit abrupt.

Following the Reviewer’s advice, we have made the suggested changes in the Introduction to enhance the coherence of our manuscript:

Page 2 (lines 76-80): Although the impact of ICIs on survival is significant, they have also been linked to autoimmune-like side effects due to their ability to stimulate the immune system. These adverse effects are often expressed in the form of neurological toxicities, hepa-totoxicity, and cardiotoxicity [28-30]. Therefore, we need a better understanding of the molecular mechanisms underlying therapeutic response.

Additionally, we added the following references to support our claims in the manuscript:

  1. Vogrig, A.; Muniz-Castrillo, S.; Farina, A.; Honnorat, J.; Joubert, B. How to diagnose and manage neurological toxicities of immune checkpoint inhibitors: an update. J Neurol 2022, 269, 1701-1714, doi:10.1007/s00415-021-10870-6.
  2. Remash, D.; Prince, D.S.; McKenzie, C.; Strasser, S.I.; Kao, S.; Liu, K. Immune checkpoint inhibitor-related hepatotoxicity: A review. World J Gastroenterol 2021, 27, 5376-5391, doi:10.3748/wjg.v27.i32.5376.
  3. Chen, R.; Zhou, M.; Zhu, F. Immune Checkpoint Inhibitors Related to Cardiotoxicity. J Cardiovasc Dev Dis 2022, 9, doi:10.3390/jcdd9110378.

  1. I would also suggest to choose either TIME or TME.

According to the Reviewer’s suggestion, we revised the manuscript and now the term TME is consistently used to refer to the “Tumor Microenvironment” throughout the text.

  1. Regarding the quality of English Language, I would suggest to tone down a little bit expressions such as "realm" "maestro" "sanguine".

We thank the Reviewer for the insightful feedback. Prompted by this, we thoroughly edited the manuscript, so that such expressions are omitted. We believe that these revisions enhance the readability and clarity of the manuscript.

The authors wish to thank the Reviewers for their constructive comments that led to the improvement of the current manuscript.
